# Impact of the HPV Vaccine on Oral HPV Infections in Indigenous Australian Adults

**DOI:** 10.3390/vaccines13070685

**Published:** 2025-06-26

**Authors:** Xiangqun Ju, Lucy Lockwood, Sneha Sethi, Joanne Hedges, Lisa Jamieson

**Affiliations:** Australian Research Centre for Population Oral Health (ARCPOH), Adelaide Dental School, University of Adelaide, Adelaide 5000, Australia; lucy.lockwood@student.adelaide.edu.au (L.L.); sneha.sethi@adelaide.edu.au (S.S.); joanne.hedges@adelaide.edu.au (J.H.); lisa.jamieson@adeelaide.edu.au (L.J.)

**Keywords:** Human Papilloma Virus, Indigenous Australian, knowledge and awareness, HPV vaccine

## Abstract

Background/Objectives: The HPV vaccine is highly effective and safe in preventing HPV infection. This study explored the relationship between HPV vaccination, HPV knowledge and awareness, and oral HPV infection prevalence among Indigenous Australian adults. Methods: Data were collected from a large convenience sample in South Australia in 2018–19, with annual follow-ups through 2022–23. The primary outcome was oral infection with HPV types 6, 11, 16, 18, 31, 33, 45, 52, or 58. The main exposure was HPV vaccination uptake status, which was categorised as unvaccinated, partially vaccinated (1–2 doses), or fully vaccinated (3 doses). Covariates included sociodemographic factors, general and sexual health behaviours, and HPV knowledge scores (HPV-KT). Risk ratios (RRs) for oral HPV infection were estimated using Poisson regression models. Results: Among the 1006 participants who completed at least one questionnaire and oral HPV test by 24 months, 81% were unvaccinated, 13% partially vaccinated, and 7% fully vaccinated. Fully vaccinated individuals had the highest HPV-KT scores (mean: 3.4) and the lowest oral HPV prevalence (5%). After adjusting for covariates, unvaccinated participants had a 1.08 times higher risk of oral HPV infection (95% CI: 1.00–3.11) compared to those fully vaccinated. Conclusions: Full HPV vaccination (three doses) is associated with lower oral HPV infection and greater HPV knowledge. The protective effect appears stronger than for partial vaccination or no vaccination, underscoring the importance of completing the full vaccine series to reduce oral HPV burden.

## 1. Introduction

In recent years, oral Human Papillomavirus (HPV) infection has been the main cause of oropharyngeal cancer (OPC) in Western developed countries [1,2,3,4]. HPV genotypes are classified into two main categories: low risk (lr-HPV) and high risk (hr-HPV) based on their oncogenic potential [5]. HPV 6 and HPV 11 are lr-HPV that are primarily linked to Recurrent Respiratory Papillomatosis (RRP), a serious condition characterised by wart-like growths in the respiratory tract that can impair voice and obstruct the airway [6,7]. High-risk types, particularly HPV-16 and HPV-18, are strongly associated with various cancers, including oral and oropharyngeal cancers (OPCs) [8,9], such as approximately 95% of HPV-positive oropharyngeal tumours in the United States [10], and in 94% of such tumours among males in Australia [11]. Worldwide, due to systemic health inequalities and disparities in access to healthcare services [12], the incidence of OPCs for Indigenous populations is higher than that for non-Indigenous populations. In Australia, Aboriginal and Torres Strait Islander peoples (hereafter respectfully referred to as ‘Indigenous’), who make up 3.3% of the population [13], have a higher prevalence of oral HPV infection compared to non-Indigenous Australians.

The HPV vaccine is highly effective and safe in preventing HPV infection if one has not been exposed to HPV infection [14]. The most common one currently is the nine-valent HPV vaccination (a Gardasil^®^9 vaccination) [15], which prevents infection caused by HPV types 6, 11, 16, 18, 31, 33, 45, 52, and 58. Australia was the first country to use the HPV vaccine, beginning it for girls in 2007 and extending it to boys in 2013. The program is mainly delivered through schools, targeting Year 7 students aged 12–13 years. Those who miss the school-based single-dose vaccination are eligible to receive the vaccine free of charge up to age 26 years, with a three-dose schedule [15,16]. In 2016, national HPV vaccination coverage among Indigenous Australian adolescents was high, with over 80% of both girls and boys receiving at least one dose. However, completion rates for the full vaccination course remained low [17].

The HPV vaccine is effective in preventing genital HPV infections [18,19]; however, limited research has explored the association between HPV vaccination and oral HPV infection [20]. This study aimed to examine the interrelationship between HPV vaccination, knowledge and awareness of HPV, and the prevalence of oral HPV infection. Additionally, it sought to estimate the potential impact of HPV vaccination on oral HPV infection. We hypothesised that HPV vaccination reduces the risk of oral HPV infection among Indigenous Australian adults.

## 2. Materials and Methods

This study followed the STROBE (Strengthening the Reporting of Observational Studies in Epidemiology) guidelines for transparent and comprehensive reporting.

### 2.1. Study Design and Data Collection

A longitudinal cohort study was conducted using data from a large convenience sample of Indigenous Australian adults in South Australia. The initial data collection took place in 2018–19 (n = 1011), with follow-up data collected from the same participants in 2020–21 (n = 743) and 2022–23 (n = 803). This study had the oversight of Aboriginal Community Controlled Health Organisations (ACCHOs) and an Indigenous Reference Group (IRG), and participants aged 18+ years were recruited through both ACCHOs and the IRG. Loss to follow-up included participants who had travelled interstate, were incarcerated, withdrew from the study (due to COVID-19 restrictions or other reasons), or had passed away [21].

### 2.2. Ethics Approval and Consent to Participate

The study received ethical approval from the University of Adelaide Human Research Ethics Committee (H-2016–246) and the Aboriginal Health Council of South Australia (04–17-729). All participants provided written informed consent.

### 2.3. Variables

#### 2.3.1. Outcome

The outcome variable was the prevalence of oral HPV infections, which was divided into the four subgroups: (1) non-vaccine types of oral HPV infections (including 44 identified HPV types: 3, 7, 10, 13, 26, 30, 32, 34, 35, 39, 40, 42, 44, 51, 53, 54, 56, 59, 62, 66, 67, 68, 69, 72, 73, 76, 81, 82, 84, 87, 89, 90, 103, 106, 107, and 182) [21]; (2) hr-HPV infection (including 18 identified hr-HPV types 16, 18, 26. 31, 33, 35, 39, 45, 51, 52, 53, 56, 58, 59, 66, 68, 73, and 82); (3) nine-valent vaccine type oral HPV infection (Gardasil^®^9 vaccination-related HPV infection: HPV types 6, 11, 16, 18, 31, 33, 45, 52, and 58); and (4) HPV types 16 and 18 infection.

HPV types were detected through DNA analysis of saliva samples collected at baseline, and at 12- and 24-month follow-up. Details have been reported in our previous publications [21,22]. Briefly, all saliva samples underwent β-globin PCR testing using PCO3/4 primers to confirm adequate human cellular content and to exclude the presence of PCR inhibitors. Samples with β-globin negative were considered invalid. HPV detection was performed using a nested PCR method with MY09/11 and GP5+/6+ primers, enabling the identification of a wide range of mucosal HPV types. This included high-risk genotypes such as HPV-16 and HPV-18, which are known for their oncogenic potential in mucosal tissues [22].

#### 2.3.2. Exposure

The exposure variable was HPV vaccination uptake status. This variable combined the baseline and 12-month follow-up questions. At baseline, this variable was assessed by the question ‘Have you ever received a vaccination for HPV?’; and at 12-month follow-up, this study was identified from responses to ‘Which of the following statements best describes whether you have had the HPV vaccine: (1) Yes, I had 1, 2, or 3 dose(s) of the HPV vaccine; (2) Yes, I have, but I am not sure how many doses I received, (3) No, I was offered the HPV vaccine, but I did not have it, (4) No, I have never been offered the HPV vaccine’. Then, HPV vaccination uptake status was categorised as ‘Had, one or two dose(s)’, ‘Had, three doses’, or ‘No’.

#### 2.3.3. Covariates

Covariates included social demographic characteristics, health and health-related behaviours, sexual behaviours, and HPV knowledge tool (HPV-KT) scores.

Sociodemographic characteristics included age, sex, residential location, highest educational level, and household income. Health and health-related behaviours included self-rated general and oral health (‘Excellent/Very good/Good’ or ‘Fair/Poor’), tobacco smoking statuses (‘Current smoker’, ‘Ex-smoker’ and ‘Never smoked’), and recreational drug use (‘Currently use’, ‘Don’t know but used to’ and ‘Never used’). Sexual behaviours included had passionately kissed (‘≥4’ vs. ‘<4’), ever given/received oral sex (‘Yes’ vs. ‘No’), given/received oral sex age (‘<16’ vs. ‘≥16’), number of oral sex given/received (‘>3’ vs. ‘≤3’), had sexual intercourse (‘Yes’ vs. ‘No’), had sexual intercourse age (‘<16’ vs. ‘≥16’), number of sexual intercourse (‘≥4’ vs. ‘<4’), and current relationship status (‘Stable and long’, ‘Short-term’, or ‘Single’). The HPV-KT assessed three domains, which included (1) general knowledge about HPV, (2) knowledge of HPV testing, and (3) knowledge of HPV vaccination [23]. HPV knowledge was assessed using the 7-item HPV-KT scale, with scores ranging from 0 to 7; higher scores reflected greater knowledge.

### 2.4. Statistical Analysis

Statistical analysis involved both descriptive and multivariate approaches.

The analysis began with univariate statistics to summarise exposure and covariates distributions, the prevalence of oral HPV infection, and mean HPV-KT scores, including 95% confidence intervals (CIs), stratified by HPV vaccination status. For multivariate analysis, logistic regression models using Poisson distribution estimates were used to assess unadjusted and adjusted risk ratios (RRs) and 95% Cis. Exposure and covariates were added to the multivariable models in four stages: Model 1 was a crude model for exposure; Model 2 included sociodemographic factors; Model 3 added health status and health-related behaviours; and Model 4 (the full model) incorporated sexual behaviour variables.

The data were analysed using the SAS software version 9.4 (SAS 9.4, SAS Institute Inc., Cary, NC, USA).

## 3. Results

The analyses included the 1006 Indigenous Australian adults who completed questionnaires at baseline and/or at the 12-month follow-up and who had at least one oral HPV test by the 24-month follow-up.

### 3.1. Sample Characteristics

Table 1 presents the sample characteristics and their associations with HPV vaccine status among Indigenous Australian adults. A greater proportion of participants were over 37 years of age (52%), female (67%), residing in non-metropolitan areas (63%), had a highest education level of high school or less (68%), received income from Centrelink (76%), currently smoked tobacco (69%), had never used non-tobacco smoking products (69%), and had never used recreational drugs (46%). Over 85% of participants did not have tonsillectomy, and more than 75% and 65% rated their general and oral health as ‘excellent,’ ‘very good,’ or ‘good.’ Approximately 65% of participants had passionately kissed more than four people, and around 70% had given and/or received oral sex. A majority of participants reported engaging in oral sex at age 16 years or older (approximately 70%), with over half (51%) having had more than three sexual partners. Additionally, 95% had experienced sexual intercourse, 59% had their first sexual intercourse at age 16 years or older, and 63% reported having had sexual intercourse more than four times. Just over half (51%) indicated that their current relationship status was stable and long-term.


vaccines-13-00685-t001_Table 1Table 1Sample characteristics and their associations with HPV vaccine status among Indigenous Australian adults.


HPV Vaccines


Nonvaccinated(n = 812)One/Two Doses(n = 129)Three Doses(n = 65)

N (%)% (95% CI)% (95% CI)% (95% CI) Total  1006 (100)80.7 (78.3–83.2)12.8 (10.8–14.9)6.5 (4.9–8.0)Sociodemographic characteristics    Age group (Years)≥37 526 (52.3)94.5 (92.5–96.4)4.9 (3.1–6.8)0.6 (0.0–1.2)<37480 (47.7)65.6 (61.4–69.9)21.5 (17.8–25.1)12.9 (9.9–15.9)GenderMale336 (33.4)91.1 (88.0–94.1)7.1 (4.4–9.9)1.8 (0.4–3.2)Female670 (66.6)75.5 (72.3–78.8)15.7 (12.9–18.4)8.8 (6.7–11.0)LocationNon-metropolitan630 (62.7)79.7 (76.5–82.8)14.4 (11.7–17.2)5.9 (4.0–7.7)Metropolitan374 (37.3)82.4 (78.5–86.2)10.2 (7.1–13.2)7.5 (4.8–10.2)Education LevelHigh school or less676 (68.1)79.9 (76.9–82.9)14.1 (11.4–16.7)6.1 (4.3–7.9)TAFE ^1^/Trade or over316 (31.9)82.9 (78.8–87.1)10.1 (6.8–13.5)7.0 (4.2–9.8)IncomeCentrelink ^2^754 (76.0)80.1 (77.3–83.0)13.1 (10.7–15.5)6.8 (5.0–8.6) Job238 (24.0)82.4 (77.5–87.2)12.6 (8.4–16.8)5.0 (2.3–7.8)Health and related behaviours    Smoking statusCurrent Smoker566 (59.5)80.7 (77.5–84.0)14.0 (11.1–16.8)5.3 (3.5–7.1) Ex-smoker113 (11.9)81.4 (74.2–88.6)13.3 (7.0–19.5)5.3 (1.2–9.5) Never smoker 272 (28.6)80.1 (75.4–84.9)11.4 (7.6–15.2)8.5 (5.1–11.8)Non-tobacco SmokingCurrently116 (12.2)81.9 (74.9–88.9)12.1 (6.1–18.0)6.0 (1.7–10.4) Used182 (19.1)76.9 (70.8–83.1)16.5 (11.1–21.9)6.6 (3.0–10.2) Never655 (68.7)81.1 (78.1–84.1)12.2 (9.7–14.7)6.7 (4.8–8.6)Recreational Drug UseCurrently207 (20.9)78.7 (73.2–84.3)15.9 (10.9–20.9)5.3 (2.3–8.4) Used332 (33.5)78.0 (73.5–82.5)15.4 (11.5–19.2)6.6 (3.9–9.3) Never453 (45.7)83.4 (80.0–86.9)9.7 (7.0–12.4)6.8 (4.5–9.2)Ever had tonsils outNo804 (86.6)79.7 (76.9–82.5)13.6 (11.2–15.9)6.7 (5.0–8.4)Yes124 (13.4)79.8 (72.8–86.9)13.7 (7.6–19.8)6.5 (2.1–10.8)Self-rated general healthFair/poor221 (22.4)85.5 (80.9–90.2)10.4 (6.4–14.4)4.1 (1.5–6.7)Excellent/very good/good767 (77.6)79.3 (76.4–82.1)13.6 (11.1–16.0)7.2 (5.3–9.0)Self-rated oral healthFair/poor329 (33.6)80.5 (76.3–84.8)12.8 (9.2–16.4)6.7 (4.0–9.4)Excellent/very good/good650 (66.4)80.8 (77.7–83.8)12.8 (10.2–15.3)6.5 (4.6–8.4)Sexual behaviours     Had kissed passionately≥4590 (65.0)78.6 (75.3–82.0)13.6 (10.8–16.3)7.8 (5.6–10.0)<4317 (35.0)83.0 (78.8–87.1)12.0 (8.4–15.6)5.0 (2.6–7.5)Ever given/received oral sexYes629 (69.4)78.5 (75.3–81.8)13.7 (11.0–16.4)7.8 (5.7–9.9)No277 (30.6)83.8 (79.4–88.1)11.6 (7.8–15.3)4.7 (2.2–7.2)Given/received oral sex age<16192 (30.9)80.7 (75.1–86.3)11.5 (6.9–16.0)7.8 (4.0–11.6)≥16430 (69.1)77.4 (73.5–81.4)14.9 (11.5–18.3)7.7 (5.2–10.2)Number of oral sex given/received>3317 (50.8)78.2 (73.7–82.8)13.6 (9.8–17.3)8.2 (5.2–11.2)≤3307 (49.2)78.5 (73.9–83.1)14.0 (10.1–17.9)7.5 (4.5–10.4)Had sexual intercourseYes848 (94.7)80.5 (77.9–83.2)12.6 (10.4–14.9)6.8 (5.1–8.5)No47 (5.3)72.3 (59.5–85.2)19.1 (7.9–30.4)8.5 (0.5–16.5)Had sexual intercourse at age<16345 (41.1)80.9 (76.7–85.0)12.8 (9.2–16.2)6.4 (3.8–9.0)≥16494 (58.9)80.4 (76.9–83.9)12.6 (9.6–15.5)7.1 (4.8–9.4)Number of sexual intercourse≥4529 (63.4)80.5 (77.1–83.9)12.3 (9.5–15.1)7.2 (5.0–9.4)<4306 (36.6)80.1 (75.6–84.6)13.4 (9.6–17.2)6.5 (3.8–9.3)Current relationship statusStable and long458 (50.7)80.3 (76.7–84.0)13.8 (10.6–16.9)5.9 (3.7–8.1)Short-term48 (5.3)70.8 (58.0–83.7)18.8 (7.7–29.8)10.4 (1.8–19.1)Single398 (44.0)81.7 (77.8–85.5)11.1 (8.0–14.1)7.3 (4.7–9.8)Notes: ^1^: Technical and Further Education, which provides training for vocational occupations; ^2^: Welfare support payments.


### 3.2. HPV Vaccine Status

A total of 81% of participants had never received the HPV vaccine, while 13% had received one to two doses, and 7% had completed all three doses. Similar trends were noted across all covariates and their corresponding subgroups. The highest percentage of individuals who had not received the HPV vaccine was observed in the older age group (≥37 years), at 95%. The highest percentage of individuals who had received one or two doses of the HPV vaccine (22%) and those who had completed all three doses (13%) were observed in the younger age group (<37 years).

### 3.3. HPV Vaccination Status and Oral HPV Infection

Figure 1 illustrates the association between HPV vaccination status and oral HPV infection. For non-vaccine types of oral HPV infections, prevalence was the highest among individuals who had received one or two vaccine doses (47%), followed by those who had completed all three doses (45%), with the lowest prevalence observed among those who were unvaccinated (43%). In contrast, for oral high-risk HPV (hr-HPV) infection, the highest prevalence was seen in individuals who had received all three doses (12%), followed by the unvaccinated group (11%), while the lowest prevalence occurred among those with one or two doses (10%). A consistent pattern was observed for vaccine-related oral HPV infections and hr-HPV 16/18 infections: the highest prevalence was among individuals who had never received the HPV vaccine (7% and 5%, respectively), followed by those who had received one or two doses (6% and 4%), with the lowest prevalence among those who had completed all three doses (5% and 3%). These findings support the effectiveness of HPV vaccination.


Figure 1Oral HPV infection by HPV vaccine status.
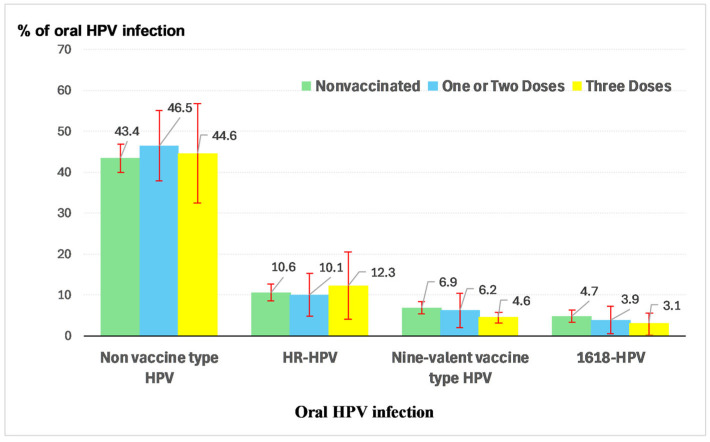



### 3.4. HPV Vaccination Status and HPV-KT Scores

Figure 2 presents the relationship between HPV-KT scores and HPV vaccination status. The mean HPV-KT score was the highest among individuals who had received all three doses of the HPV vaccine (3.4), followed by those with one or two doses (3.0), and the lowest among those who had not received the vaccine (2.1).

### 3.5. Risk Ratios (RRs) for Oral HPV Infection

Table 2 presents the association between oral HPV infection related to the nine-valent vaccine and HPV vaccination status, along with other risk factors. Participants who had not received the HPV vaccine or had received only one or two doses had approximately 1.4 times (RR = 1.39, 95% CI: 1.01–4.51) and 1.3 times (RR = 1.34, 95% CI: 1.02–5.06) higher risk, respectively, of having oral HPV infection compared to those who had completed all three doses of the HPV vaccine (Model 1). After adjusting for sociodemographic characteristics, health-related behaviours, and sexual behaviours (Model 4), participants who had not received the HPV vaccine had approximately 1.1 times higher risk of having oral HPV infection (RR = 1.08, 95% CI: 1.00–3.11) compared to those who had completed all three doses of the vaccine. Although the difference between receiving one or two doses and receiving three doses was not statistically significant (RR = 1.06, 95% CI: 0.89–3.83), the wide confidence interval indicates uncertainty, and the potential effect of completing all three doses cannot be ruled out.


vaccines-13-00685-t002_Table 2Table 2Models for nine-valent vaccine-related oral HPV infection (RRs and 95% CI).

Model 1Model 2Model 3Model 4
RR (% CI)PR (% CI)RR (% CI)RR (% CI)HPV vaccination statusNonvaccinated  1.39 (1.01–4.51) 1.39 (1.02–4.68)1.19 (1.02–4.15)1.08 (1.00–3.11)One/Two doses1.34 (1.02–5.06)1.59 (1.04–6.03)1.17 (0.98–4.81)1.06 (0.89–3.83)Three dosesrefrefrefrefSociodemographic characteristics     Age group (Years)≥37 0.97 (0.58–1.63)1.10 (0.60–2.01)1.39 (0.66–2.93)<37 refrefrefGenderMale 1.37 (0.82–2.29)1.41 (0.78–2.55)1.52 (0.71–3.27)Female refrefrefLocationNon-Metropolitan0.29 (0.17–0.48)0.28 (0.16–0.52)0.27 (0.12–0.59)Metropolitan refrefrefEducation LevelHigh school or less0.68 (0.41–1.15)0.60 (0.34–1.07)0.54 (0.27–1.09)Trade or over refrefrefIncomeCentrelink 1.51 (0.81–2.79)1.44 (0.74–2.81)1.42 (0.66–3.07)Job refrefrefHealth and related behaviours     Smoking statusCurrent Smoker  0.72 (0.36–1.42)0.75 (0.32–1.78)Ex-smoker  1.14 (0.49–2.64)0.83 (0.29–2.41)Never smoker   refrefNon-tobacco SmokingCurrently  1.14 (0.59–3.37)1.11 (0.36–3.44)Used  1.40 (0.71–2.75)1.43 (0.67–3.07)Never  refrefRecreational Drug UseCurrently  0.87 (0.36–2.09)1.07 (0.36–3.21)Used  1.28 (0.66–2.50)1.45 (0.62–3.40)Never  refrefEver had tonsils outNo  0.82 (0.40–1.71)0.67 (0.29–1.52)Yes  refrefSelf-rated general healthFair/poor  1.31 (0.64–2.68)1.60 (0.67–3.86)Excellent/very good/good refrefSelf-rated oral healthFair/poor  0.89 (0.48–1.65)0.80 (0.39–1.65)Excellent/very good/good refrefSexual behaviours     Had kissed passionately≥4   1.57 (0.45–5.45)<4   refEver given/received oral sexYes   1.07 (0.40–2.83)No   refGiven/received oral sex age<16   1.30 (0.51–3.31)≥16   refNo of oral sex given/received>3   0.94 (0.37–2.41)≤3   refHad sexual intercourseYes   0.78 (0.28–2.15)No   refHad sexual intercourse at age<16   0.85 (0.50–1.47)≥16   refNo. of sexual intercourse with≥4   1.90 (1.02–3.54)<4   refCurrent relationship statusStable and long   0.83 (0.25–2.67)Short-term   0.64 (0.14–2.99)Single   refNotes: RR: risk ratio; Model 1: crude model; Model 2: HPV vaccine status plus adjusting for sample characteristics; Model 3: Model 2 plus adjusting for health and health-related behaviours; Model 4: full model—Model 3 plus adjusting for sexual behaviours.


As the associations between other subgroups of oral HPV infection and HPV vaccination status were not statistically significant, they are not reported in this paper.

## 4. Discussion

Our findings support the hypothesis that HPV vaccination lowers the risk of oral HPV infection among Indigenous Australian adults. The vaccine was most effective among those who received all three doses, showing a stronger protective effect compared to individuals who received only one or two doses or were unvaccinated. Additionally, higher HPV knowledge scores and a lower prevalence of oral HPV infection were observed in this population. The data also indicate that a higher proportion of younger (<37 years) and female Indigenous Australian adults had received at least one dose of the HPV vaccine.

Our findings align with previous studies [24], showing that the prevalence of oral HPV types 6, 11, 16, 18, 31, 33, 45, 52, and/or 58 was significantly lower among individuals who received all three vaccine doses compared to those who were unvaccinated (5% vs. 7%). This difference was particularly notable for HPV types 16 and 18, with a prevalence of 3% among vaccinated individuals versus 5% among the unvaccinated. These results are especially important as they demonstrate the effectiveness of the HPV vaccine in reducing the risk of cancers, especially for oropharyngeal cancers (OPCs). At the same time, the potential protective effect of receiving one or two doses of the HPV vaccine should not be overlooked, as the prevalence of infection in this group was also lower than among unvaccinated individuals, although the difference was not statistically significant.

Our study found that three-dose HPV vaccine coverage among Indigenous Australian adults was well below the globally recommended threshold of 90% [25], with coverage as low as 0.6% among the older age group (≥37 years) and 13% among the younger age group (<37 years). Possible reasons included: (1) older individuals were not eligible for the HPV vaccine during their youth, as the national HPV vaccination program in Australia did not commence until 2007; (2) Australia’s HPV vaccination program is school-based, and evidence indicates lower school attendance rates among Indigenous students [16]; and (3) a lack of culturally appropriate delivery of vaccination information and services [26].

### Strengths and Limitations

The strengths of this study include: (1) a large and representative sample of the broader South Australian Indigenous population (n = 1006), with findings that may be relevant to other Indigenous communities in comparable socio-economic contexts globally; and (2) meaningful engagement with Indigenous Australian communities through established partnerships and the active involvement of the study’s Indigenous Reference Group. The limitations of this study include: (1) potential lack of representativeness, as the sample had a higher proportion of females, individuals residing in non-metropolitan areas, those with welfare-based income, and current smokers compared to the broader Indigenous population [27]; (2) loss to follow-up at the 12- and 24-month time points, which may have introduced bias in the estimates; (3) the combination of reported sexual behaviours (i.e., giving and receiving oral sex) may limit accuracy, as previous research has indicated that the highest risk of oral HPV infection may be associated specifically with giving cunnilingus [28]; (4) our study lacked data on the precise timing of vaccine doses, limiting our ability to evaluate schedule-specific effects on immunogenicity and protection. Future research should incorporate more detailed vaccination timing data to enable such assessments; and (5) some detected HPV DNA may represent transient deposition rather than established productive infections. However, we employed validated HPV DNA detection methods with high sensitivity and specificity, which reduce the likelihood of contamination. Moreover, we did not assess markers of viral transcription or replication (e.g., E6/E7 mRNA). Future studies incorporating biomarkers of viral activity could help distinguish between transient deposition and true infection.

Despite these limitations, our findings provide further evidence that HPV vaccination coverage remains extremely low and highlight the need for policymakers to implement culturally safe educational strategies alongside a comprehensive vaccination program to increase coverage among Indigenous Australians and reduce the burden of HPV-related cancers.

## 5. Conclusions

Our findings indicate that people who received all three doses HPV vaccination had a stronger effect in reducing oral HPV infection compared to individuals who received only one or two doses or were unvaccinated. Additionally, higher HPV knowledge scores and a lower prevalence of oral HPV infection were observed in this population.

## Figures and Tables

**Figure 2 vaccines-13-00685-f002:**
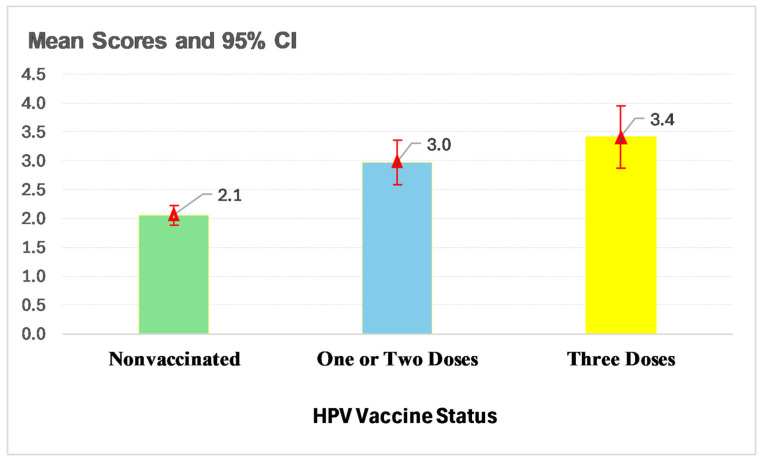
Mean scores and 95% CI of HPV-KT by HPV vaccine status.

## Data Availability

The datasets generated and/or analysed during the current study are not publicly available due to privacy issues of the participants. Data are available from the corresponding author upon reasonable request.

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
