# Peer review of "Impact of the HPV Vaccine on Oral HPV Infections in Indigenous Australian Adults"

_vaccines, 2025, doi:10.3390/vaccines13070685_

Round 1

Reviewer 1 Report

Comments and Suggestions for Authors

The overall merit of your study is related to the inequity in health for indigenous populations, the high prevalence of herpes in this group, the extremely low vaccination coverage and the need for a complete vaccination programme.

I appreciate the recognition of the strenghts and limitations of the study.

What is not obvious in your analysis is the impact of uncomplete coverage and the duration of impact? You may look at these issues in the litterature and your data analyses and mention your views in the discussion section.

Author Response

Please see the attachment,

Reviewer 2 Report

Comments and Suggestions for Authors

This paper describes an interesting study designed to determine the impact of vaccination with the 9v HPV vaccine on oral HPV prevalence in indigenous Australians.

The study is well designed but there several omissions which need to be corrected. 

  • Describe the Australian National HPV vaccination programme and the dosing schedule that the cohort understudy would/should have received. The authors show evidence that 3 doses were better than 1 or2 but not which schedule they were in.  This matters since 3 doses 1,2,and 6 or 12 months results in GMTs comparable to a 2dose schedule at01 and 6 months but not if the schedule is 0,2 months.
  • Could they separate the prevalence of non vaccine types in the different groups (FIGURE 1) rather than prevalence of all oral HPV, it would be a better reference group than the overall prevalence
  • Have the authors any comment on the possibility that many of these infections were not established productive infections but deposition?
  • The DNA detection methodology – has this been published in detail and validated

The methodology is important since the viral loads are small 

Round 2

Reviewer 1 Report

Comments and Suggestions for Authors

Thank you for your reply and amendments made

Reviewer 2 Report

Comments and Suggestions for Authors

The authors have responded satisfactorily to the review comments.